

# beadplexr: reproducible and automated analysis of multiplex bead assays

Ulrik Stervbo[1,2], Timm H. Westhoff[1] and Nina Babel[1,2]

[1] Center for Translational Medicine, Medical Clinic I, Marien Hospital Herne, University Hospital of the Ruhr-University Bochum, Herne, Germany

[2] Charité—Universitätsmedizin Berlin, corporate member of Freie Universität Berlin, Humboldt-Universität zu Berlin, and Berlin Institute of Health, Berlin-Brandenburg Center for Regenerative Therapies, Berlin, Germany

## ABSTRACT

Multiplex bead assays are an extension of the commonly used sandwich ELISA. The advantage over ELISA is that they make simultaneous evaluation of several analytes possible. Several commercial assay systems, where the beads are acquired on a standard flow cytometer, exist. These assay systems come with their own software tool for analysis and evaluation of the concentration of the analyzed analytes. However, these tools are either tied to particular commercial software or impose other limitations to their licenses, such as the number of events which can be analyzed. In addition, all these solutions are 'point and click' which potentially obscures the steps taken in the analysis. Here we present `beadplexer`, an open-source R-package for the reproducible analysis of multiplex bead assay data. The package makes it possible to automatically identify bead clusters, and provides functionality to easily fit a standard curve and calculate the concentrations of the analyzed analytes. `beadplexer` is available from CRAN and from https://gitlab.com/ustervbo/beadplexr.

## INTRODUCTION

The enzyme-linked immunosorbent assay (ELISA) is a commonly used method to determine the concentration of soluble analytes such as cytokines (*Elshal & McCoy, 2006*). The concentration of the analyte is determined from a standard curve, which is created from standard samples with known concentrations. The ELISA is a single point assay and query into several analytes can be time consuming or impossible when the sample is limited. Development in polystyrene bead preparations made it possible to construct assays that allow for query of several analytes at the same time. Similar to the ELISA, the analytes of interest are captured by a primary antibody (Fig. 1A). The captured analytes are subsequently labelled with a secondary antibody which in turn is detected with a fluorochrome conjugated tertiary antibody. The level of fluorochrome intensity is directly related to the amount of bound tertiary antibody, and therefore also to the amount of analyte present in the sample. In a multiplex bead assay, the primary antibody is fixed on a polystyrene bead, and physical properties such as size and granularity as well as fluorescent

Corresponding author
Ulrik Stervbo,
ulrik.stervbo-kristensen@charite.de

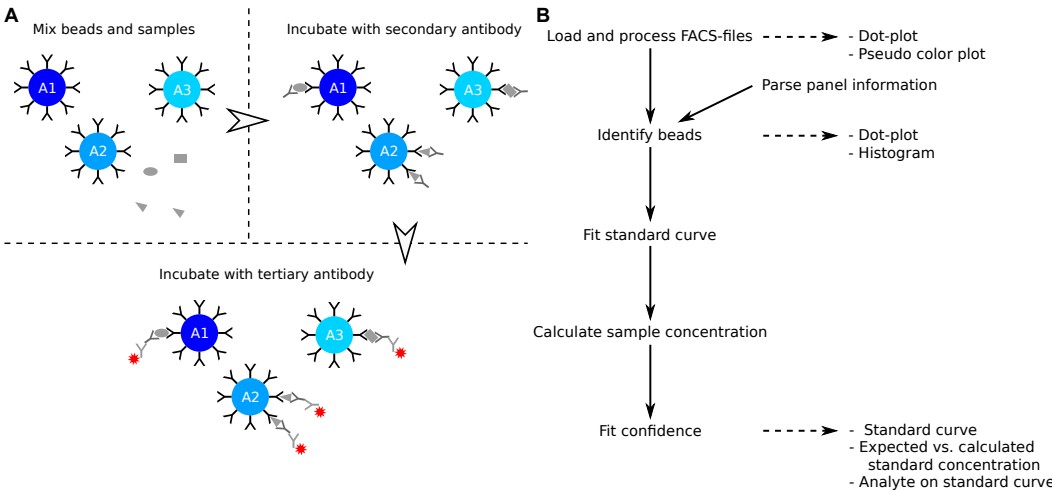

**Figure 1  Overview of assay principle and the package workflow.** (A) Schematic overview of the principle of a LEGENDplex assay. (B) Steps in analysis of a multiplex bead assay with accompanying visualizations.

colors of the beads are used to distinguish the different analytes studied. The data is usually collected using a standard flow cytometer.

The LEGENDplex system from BioLegend, the CBA system from BD Biosciences, and the MACSPlex system from Miltenyi Biotec are all bead based multiplex systems (*Morgan et al., 2004*; *Miltenyi Biotec, 2014*; *Yu et al., 2015*). The systems differ slightly in terms of physical properties and colors used, and in the number of analytes that can be simultaneously identified. The Bio-Plex system from Bio-Rad works in a similar manner as those described here, but requires a dedicated instrument and does not produce files suitable for analysis with beadplexr. The individual assays that can be analyzed with beadplexr are described in the following.

**LEGENDplex:** Beads fall into two large groups based on size and granularity—as related to the forward light scattering, FSC, and the perpendicular light scatter, SSC. Within each group, individual analytes are discriminated by the intensity of Allophycocyanin (APC) of the beads. The concentration of the analyte is related to the intensity of Phycoerythrin (PE).

**CBA:** All beads have similar size and granularity. The individual analytes are discriminated by the intensity of APC and APC-Cy7 of the bead. The concentration of the analyte is related to the intensity of PE.

**MACSPlex:** All beads have similar size and granularity. The individual analytes are discriminated by the intensity of PE and Fluorescein isothiocyanate (FITC) of the bead. The concentration of the analyte is related to the intensity of APC.

All multiplex systems come with their own analysis software. However, these solutions might come with an added price tag because of binding to a particular piece of software, or the license is valid only for a number of bead events. In this case, large data files with many bead events or repeated re-evaluation of the acquired data might result an expiration of

the license. In addition, the usability and flexibility of the analysis solutions are restricted and often impractical for experiments with a large number of samples. Currently no open source alternative exists.

Here the general usage of the `beadplexr` package for R (*R Core Team, 2018*) is introduced. It will be demonstrated how to load the files generated by the flow cytometer, identify bead populations, draw standard curves and calculate concentration of the experimental samples.

## MATERIALS & METHODS

The `beadplexr` package includes data from an unpublished "Human Growth Factor Panel (13-plex)" LEGENDplex (BioLegend) experiment performed in our laboratory. The dataset consists of eight controls samples and a serum sample from a single healthy volunteer. All samples were processed in duplicates and per manufacturer's instructions. The data was acquired on a CytoFLEX cytometer (Beckman Coulter, Brea, CA, USA). An example of a flow cytometry data file is also included in the package. We utilize these data to illustrate the functionality of the package.

The data here were analyzed with R, version 3.5.1, (*R Core Team, 2018*) and plots created with `ggplot2` (*Wickham, 2009*) and `cowplot` (*Wilke, 2017*). The workflow and examples presented here make use of or suggests the following R-packages: devtools (*Wickham, Hester & Chang, 2018*), `dplyr` (*Wickham et al., 2018*), `hexbin` (*Carr et al., 2018*), `magrittr` (*Bache & Wickham, 2014*), `purr` (*Henry & Wickham, 2018*), `stringr` (*Wickham, 2018*), and `tidyr` (*Wickham & Henry, 2018*).

## RESULTS

### Package overview

The released package can be installed from CRAN and the development version from GitLab:

```
# Installing the package ------------------------------------------

# From CRAN
install.packages("beadplexr")
# From GitLab using devtools
# install.packages("devtools")
# devtools::install_git("https://gitlab.com/ustervbo/beadplexr")
#
# Or with vignettes built
# devtools::install_git(https://gitlab.com/ustervbo/beadplexr",
# build_vignettes = TRUE)
```

The package provides several steps to extract the analyte concentration from the raw data (Fig. 1B). The functions for interacting with the data are flexible, but sensible defaults make them accessible to the novice R-user. The workflow and examples presented here are

collected in Script S1, and a more detailed workflow is presented in the package vignette. The latter can be viewed using the command `vignette("legendplex-analysis")`.

## Reading FCS-files

beadplexr works with Flow Cytometry Standard (FCS) files (*Seamer et al., 1997*), which is the usual output of a flow cytometer. The function `read_fcs()` loads the given FCS-file using the functionality provided by the Bioconductor package `flowcore` (*Ellis et al., 2017*) and performs the following steps:

1. Apply an *arcsinh* transformation of the bead channels—this natural logarithm based transformation generally performs well on all flow cytometry data (*Finak et al., 2010*). Opposed to the traditionally used *log* 10 scaling of flow cytometry data, the *arcsinh* can deal with the negative values produced by some newer digital flow cytometers
2. Remove boundary events of the size (FCS) and granularity (SSC) channels—events outside the range of the detectors are registered with the maximum value possible. These events can interfere with the clustering
3. Optionally subset the channels to contain just bead events—similar to removal of boundary events, this might improve identification of the bead clusters
4. Convert the FCS-data to a `data.frame`

```
# Reading fcs-files ------------------------------------------------
library(beadplexr)

# Get the path to the example fcs-file
.file_name <- system.file("extdata",
                "K2-C07-A7.fcs",
                package = "beadplexr")
# 'read_fcs()' requires at least a path and filoe name of the file to
# load, by identifying the required forwars  and side scatter and the
# bead property channels, only the required data is returned.
#
# The argument '.filter' takes a named list, where each element is a
# size 2 vector, giving the lower and upper cut-off for the channel
# given in the element name
.data <- read_fcs(
 .file_name = .file_name,
 .fsc_ssc = c("FSC-A", "SSC-A"),
 .bead_channels = c("FL6-H", "FL2-H"),
 .filter = list(
 "FSC-A" = c(3.75e5L, 5.5e5L),
 "SSC-A" = c(4e5L, 1e6L),
 "FL6-H" = c(7L, Inf)
 )
)
```

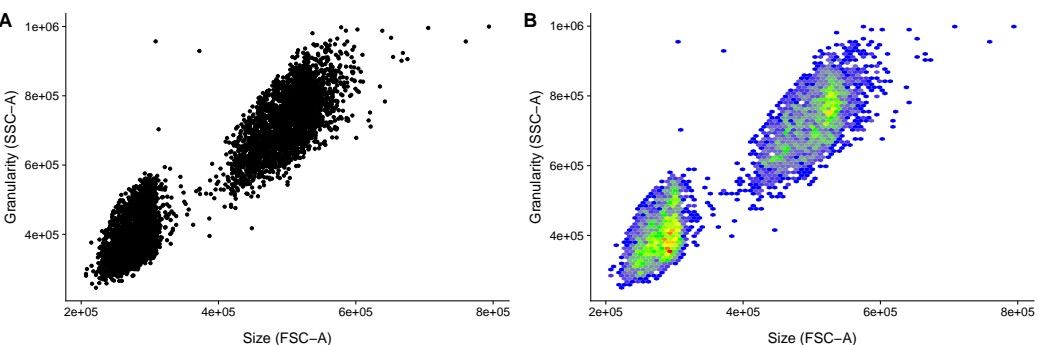

**Figure 2** **Visualization of FACS data.** Size (FSC) and granularity (SSC) can be used distinguish the two LEGENDplex bead populations. (A) Common monochrome scatter-plot created with `facs_plot(.x = "FSC-H", .y = "SSC-H", .beads = "Bead group")` on the sample 'K3-C0-1.fcs'. High density regions are obscured in this type of plots. (B) Pseudo-colored scatter -plot created with `facs_hexbin(.x = "FSC-H", .y = "SSC-H", .beads = "Bead group", .bins = 75)` on the same sample as in (A). The number of events in discrete bins is indicated by color. The coloring is according to the standard blue-green-yellow-red scheme, where blue indicates a low number of events, and red indicates a high number. The Pseudo-colored scatter -plot requires the R-package `hexbin` to be installed.

Because of the variation in detector settings between flow cytometers, it is left to the user to get the event filtering settings correct for an experiment. However, the event filtering should remain stable once established. This, of course, requires that there is no change of cytometer, and that there is no particular drift in the used cytometer. Visualizing the populations greatly helps in setting the appropriate cut-offs (Fig. 2). It is for this reason that the ggplot2 based convenience function `facs_plot()` is included.

### Naming the FCS-files

Each sample in a multiplex bead assay must have a unique and meaningful name. A later step in the workflow separates standard samples from experimental samples. The standard samples are in addition ordered in a way that calculation of dilution of standard concentrations is possible. For the dataset included in the package, 'C' followed by an integer denotes the standard (control) samples—as suggested in the LEGENDplex manual —and 'S' followed by an integer denotes the experimental samples. The different parts of the file name should be separated by a character not used in the IDs; this will make for easy parsing of the file names.

### Identification of analyte MFI

The mean fluorescence intensity (MFI) of each analyte relates directly to the concentration of the analyte in the sample (Fig. 1A). The first step to calculate the analyte concentration is to identify the bead populations representing the analytes and calculate the MFIs of these.

beadplexr makes use of structured Panel Information to provide analyte metadata such as name and start concentration for each standard sample, as well as the name of the panel, the fold dilution of the standards, and the units of the analytes. The desired Panel Information is loaded using the `load_panel()` function by passing the name or a name pattern to the function. The package itself comes with a set of LEGENDplex

Panel Information, which are documented in the help files to `load_panel`. The Panel Information file itself is in YAML format, and the `load_panel()` function can also load a Panel Information file located outside the package. The latter is useful in the cases of custom panels. The Panel Information is not required, but makes sense if the assay is repeated across several projects.

```
# Libraries -------------------------------------------------------

library(beadplexr)
library(ggplot2)
library(cowplot)
library(dplyr)
library(purrr)
library(tidyr)
library(readr)
library(stringr)

# Load data -------------------------------------------------------

data(lplex)
# Load one of the panels distributed with the package, see
# ?load_panel() for the included panels
panel_info <- load_panel(.panel_name =
                    "Human Growth Factor Panel (13-plex)")
```

Analytes of any assay system are identified using the function `identify_analyte()`, which identifies analyte clusters and assign an analyte ID to each cluster. The function takes a `data.frame` with events and a character vector giving the name of column(s) where the analytes can be discriminated. An identifier for each analyte is passed in the argument `.analyte_id`, which is simply a character vector giving the ID of the analyte. `identify_analyte()` sorts the clusters based on their centers and use this ranking to assign the analyte IDs. The order of analyte IDs given in `.analyte_id` is therefore important and must match the expected order of analytes. An optional argument is `.trim` which allows the removal events in the periphery of a cluster. The value of the argument gives the fraction of the most distant points to be removed. Distance based trimming is non-trivial since the possible numerical range depends on the detection range of the flow cytometer.

The function `identify_analyte()` interfaces several methods for unsupervised clustering, which are passed in the `.method` argument. The default clustering method is clustering large applications (`clara`) from the package `cluster` (*Maechler et al., 2017*). The method selects a number of subsets of fixed size and applies the partitioning around medoids (pam)-algorithm to each subset. The objective of the pam-algorithm is to minimize the dissimilarity between the representative of $k$ clusters and the members of each cluster (*Kaufman & Rousseeuw, 2009*). The best resulting set of medoids (cluster

centers) is that with the lowest average dissimilarity of all points in the original dataset to the medoids. Though similar to pam in algorithm type, the Base-R included kmeans works on minimizing the distance to the cluster representative (*Zaki & Wagner Meira, 2014*).

The dbscan method in the fpc package differs from clara and kmeans in that dbscan identifies clusters based local density (*Hennig, 2015*). The function requires a neighborhood size and minimum number of events in each neighborhood to evaluate whether points can be considered as belonging to a cluster (*Zaki & Wagner Meira, 2014*). If the bead populations have different local densities, there is no guarantee that the correct number of clusters will returned. This problem does not exist for Mclust from the mclust package, which fits a Gaussian mixture model using the EM-algorithm (*Scrucca et al., 2016*). This algorithm iteratively optimizes the individual parameters of *k* normal distributions (*Zaki & Wagner Meira, 2014*). This way the relationship between a cluster and a set of data points is given by a set of probability scores.

We have found that dbscan() is the best clustering method for the forward-side scatter population identification. However, it can be difficult to get the parameters *event count* and *neighborhood size* correct. The reason for this difficulty lies in the sensitivity of the method to the choice of *neighborhood size*; if it is too large clusters might be merged, and if it is too small everything might be classified as noise. In our experience, the clustering function clara() is a great all-rounder although the subsampling performed by the function can lead to slight differences between each run. Using the same value for set.seed() at the beginning of each session will alleviate this and make each run reproducible.

Different flow cytometers perform differently in terms of separation of the individual bead populations. This is due to factors such as detector settings and age of the cytometer and its light sources. The consequence is that the populations of interest might be closer together or further apart. Another consequence might be an increased in the noise of the detectors of the flow cytometer. Collectively these differences in the data constitution means that one clustering function might perform better on one dataset while be inferior on another. As with analysis of all flow cytometric data the optimal solution is a matter of taste, but the better clustering function is the one that separates the populations well, without including too much noise.

The function identify_legendplex_analyte() can be applied to each sample individually in a loop. However, it is more prudent to apply the function to all samples at the same time because the clustering decision will be identical for each sample. In addition, clustering on all the samples is 1.4 times faster than clustering on each sample individually.

```
# Identify analytes ------------------------------------------

# The function 'identify_legendplex_analyte()' used here is
# convenience around the clustering work horse 'identify_analyte'. The
# 'identify_legendplex_analyte()' identifies the bead populations
# according to size and granularity, and for each of the two
```

```
# populations the individual bead populations are identified
#
# The function requires a named list with analytes from the Panel
# Information, and a list with a list of key-value pairs giving the
# arguments for the bead identification on the forward and side
# scatter, and a list of key-value pairs giving arguments for the bead
# identification in each subpopulation in the APC channel.
#
# The argument .trim gives the fraction of events furthest from the
# centers of the groups that should be removed. The population center
# is found by a Gaussian kernel estimate. In this case we remove 1%
# and 3% of the of the events based on their distance to the group
# center.
#
# The inner lists can be named, but this is not required.
args_ident_analyte <- list(fs = list(.parameter = c("FSC-A", "SSC-A"),
                        .column_name = "Bead group",
                        .trim = 0.01),
              analyte = list(.parameter = "FL6-H",
                        .column_name = "Analyte ID",
                        .trim = 0.03))

# The FCS-data is a list of samples, which we combine before cluster
# identification.
analytes_identified <- lplex %>%
 bind_rows(.id = "Sample") %>%
 identify_legendplex_analyte(.analytes = panel_info$analytes,
                    .method_args = args_ident_analyte)
```

The analyte IDs for the "Human Growth Factor Panel (13-plex)" bead group A are A4, A5, A6, A7, A8, A10 and for group B the analyte IDs are B2, B3, B4, B5, B6, B7, B9. In this case, the beads are arranged from low to high, that is the lowest analyte ID has lowest intensity in the APC channel (Fig. 3).

This initial and crucial step of the analysis has been successfully performed with data from a CBA experiment (C McGuckin, CTIBIOTECH, Lyon, France, 2017, unpublished data) and from a MACSPlex experiment (Miltenyi Biotec, Bergisch Gladbach, Germany, 2017, unpublished data) using the function `identify_analyte()`.

With the analytes identified and the bead populations documented, the MFI of each analyte can finally be calculated. The function `calc_analyte_mfi()` gives the possibility to calculate geometric, harmonic, and arithmetic mean of the in intensity of each respective analyte reporter, such as PE in a LEGENDplex assay. Since the reporter intensities are usually log-transformed only the geometric mean is relevant, but harmonic and arithmetic mean are included to accommodate for special cases.

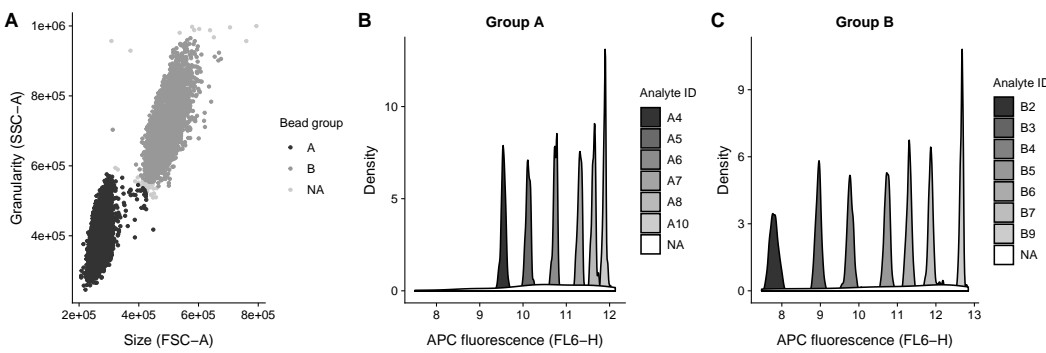

**Figure 3** **Bead identification and visualization of LEGENDplex data.** Populations identified in the sample 'K3-C0-1.fcs'. (A) Identification of the two bead populations 'A' and 'B' according to size and granularity: The two clusters were identified using `.method = clara` and noisy data points were excluded by `.trim = 0.01`. (B–C) Identification of analytes of the bead population 'A' and 'B': The 1 dimensional clusters along the APC channel were identified using `.method = clara` and noisy data points were excluded by `.trim = 0.03`. Noisy data points are assigned the group 'NA'.

```
# Calculate analyte MFI ------------------------------------------

# The mean fluorescence intensity is calculated for each sample and
# analyte. The function 'calc_analyte_mfi()' provides three ways of
# calculating the MFI: geometric, harmonic, and arithmetic mean.
analyte_mfi <- analytes_identified %>%
 filter(!is.na('Analyte ID')) %>%
 # Call 'calc_analyte_mfi()' for each sample
 group_by(Sample) %>%
 do(calc_analyte_mfi(., .parameter = "FL2-H",
            .column_name = "Analyte ID",
            .mean_fun = "geometric")) %>%
 # Later we will fit the standard curve on a log-log scale, so we
 # transform here
 mutate('FL2-H' = log10('FL2-H'))
```

## CALCULATION OF STANDARD AND EXPERIMENTAL SAMPLES

The calculation of the concentration of the analytes of the experimental samples requires two steps:

1. Create a standard curve by fitting a model to the MFI of the standard analytes and their known concentrations
2. Estimate the concentration of each sample analyte from the fitted model.

The samples in the dataset included in the package can be distinguished by the presence of 'C' or 'S', respectively. The sample type indicating letter is then followed by one or

more integers. Using this naming scheme, it is easy to separate standard samples from the experimental samples. It is also easy to order the standard samples for concentration assignment. In this case the naming scheme suggested in the LEGENDplex assay protocol is followed: 7 indicates the highest concentration of the standard analyte, 1 indicates the lowest concentration, and 0 indicates blank.

The order of the standard samples is crucial for the function `calc_std_conc()` to correctly calculate the concentration of an analyte in each standard sample. The function requires a vector which gives the order of the standard samples, a start concentration for the analyte, and a dilution factor. The standard samples are ordered numerically from high to low and assigned a standard concentration, such that the first sample is given the start concentration and the second to last sample the lowest concentration, and the very last sample the concentration 0, as this is assumed to be for background measurement.

The start concentration is stored in the Panel Information for each analyte separately, as the start concentration might differ from analyte to analyte. The dilution factor is also given in the Panel Information. It will always be the same for all standard analytes and is usually 4, meaning that the concentration of each standard analyte is 4 times lower than the previous concentration. This generally gives a good range of standard concentrations.

```
# Helper function to extract the sample number ----------------------

#' Cast sample ID to numeric
#'
#' @param .s A string with the sample ID pattern to be cast
#' @param .pattern A string giving the pattern
#'
#' @return
#' A numeric
#'
as_numeric_sample_id <- function(.s, .pattern = c("C[0-9]+", "S[0-9]+")){
  .pattern <- match.arg(.pattern)

  # Extract the pattern defined just above, remove the first element, and
  # cast to a numeric
  .s %>%
    str_extract(.pattern) %>%
    str_sub(start = -1L) %>%
    as.numeric()
}

# Split in standard and sample ------------------------------------

# We need to fit a standard curve on the standard samples, and use
# this curve to calculate the concentration of the experimental
```

```
# samples. Here we split the data set in two: one with the standard
# samples and one with the experimental samples.
#
# We need to order the standard samples from high to low in order to
# calculate the concentration of the analytes in the standard sample.
# Incorporating the information into the sample name in terms of an
# easily parsable pattern is a good practice.

# All standard samples have the pattern C[number]
standard_data <-
 analyte_mfi %>%
 ungroup() %>%
 filter(str_detect(Sample, "C[0-9]+")) %>%
 mutate(`Sample number` = as_numeric_sample_id(Sample,
 . pattern = "C"))%>%
 select(-Sample)

# All non-standards are experimental samples... we could also filter
# on S[number]
experiment_data <- analyte_mfi %>%
 ungroup() %>%
 filter(!str_detect(Sample, "C[0-9]+")) %>%
 mutate(`Sample number` = as_numeric_sample_id(Sample,
 . pattern = "S"))%>%
 select(-Sample)

# To the standard data we have to add additional information such the
# start concentration of each standard analyte and the dilution
# factor, as well as as the analyte names (analyte IDs by themselves
# do not make much sense).
#
# The concentration of the standard samples is calculated using
# `calc_std_conc()`, which take a vector of sample numbers for
# ordering, a start concentration and a dilution factor.
standard_data <- standard_data %>%
 left_join(as_data_frame_analyte(panel_info$analytes),
 by = "Analyte ID") %>%
 rename(`Analyte name` = name) %>%
 group_by(`Analyte ID`, `Analyte name`) %>%
 mutate(
  Concentration = calc_std_conc(
    `Sample number`,
    concentration,
```

```
      .dilution_factor = panel_info$std_dilution
  )
 ) %>%
 # Later we will fit the standard curve on a log-log scale, so we
 # transform here
 mutate(Concentration = log10(Concentration)) %>%
 select(-concentration, -'Bead group')
```

The next step is to fit a standard curve for each analyte. With the standard curve we can calculate the concentration of the experimental samples (the purpose of the initial work), we can check the quality of the measurements and the standard curve, and plot the experimental samples on the standard curve (beadplexr provides easy access to all of this). The latter is to allow for visual verification that the experimental samples are within the linear part of the standard curve.

However, in each case we need to ensure that the correct standard curve is used with the correct experimental data, which means we have to juggle at least three structures: A data.frame with the standard data, a data.frame with the experimental sample data, and the models for each analyte (probably a list). It quickly becomes tedious to ensure that everything is in the correct order—and it is most certainly error prone. To circumvent this, we can use the nest() and its inverse unnest() functions of the tidyr package. nest() relies the fact that a data.frame in R is in fact a list, and uses this to pack a data.frame into a single cell of a data.frame.

```
# Nest and combine standard and experimental data --------------------

# Nested data.frames is a great way of combining and working with
# complex data structures.
#
# First we pack all the standard data in to a data.frame with a set of
# data.frames
standard_data <- standard_data %>%
 nest(-'Analyte ID', .key = "Standard data")

# The the same for all the experimental data
experiment_data <- experiment_data %>%
 nest(-'Analyte ID', .key = "Experimental data")

# Since both structures are data.frames we can easily combine them
plex_data <- inner_join(standard_data, experiment_data,
               by = "Analyte ID")
```

With everything in a neatly arranged data.frame we can now focus on the actual task at hand, namely calculation of the standard curve for each analyte. For this we use the function fit_standard_curve(), which interfaces the drm() function from the drc

package (*Ritz et al., 2015*). The drm() function specializes in fitting various biological response-models, and the drc package provides several response-models, such as the four- and five-parameter log–logistic model. fit_standard_curve() is designed to be used in the piped workflow, and takes a data.frame with MFIs and concentrations and returns the model as a drc object. The four-parameter log–logistic model is widely used in analysis of ELISA data. Since the five-parameter model yields better fits, because of the increased flexibility, this is the default function (*Gottschalk & Dunn, 2005*).

```
# Calculate standard curves -------------------------------------

# For each of the analytes we calculate the standard curve. Working
# with nested data.frames means that we have to loop over each row to
# calculate the standard curve using the data.frame in "Standard data"
#
# When clustering is performed with mclust, the package mclust is
# loaded in the background (an unfortunate necessity). The mclust
# package also has a function called 'map', so an unlucky side effect
# of clustering with mclust, is that we need to be specify which map
# function we use.

plex_data <- plex_data %>%
 group_by('Analyte ID') %>%
 mutate('Model fit' = purrr::map('Standard data',
                      fit_standard_curve,
                      .parameter = "FL2-H"))
```

We can plot the standard curve using the built in plot_std_curve() function (Fig. 4A). With the standard curve created we can calculate the concentrations of the experimental samples using the function calculate_concentration(), which requires a data.frame with the MFIs in a column, and the fitted model. It can be helpful to apply calculate_concentration() to the standard samples, as this can be used to verify that the standard measurements were all fine, and that the estimation of the sample concentrations therefore is trustworthy.

After calculating the concentrations we can plot the known standard concentrations versus the estimated standard concentrations using the function plot_target_est_conc() (Fig. 4B) and visualize where the samples fall on the standard curve with plot_estimate() (Fig. 4C).

```
# Calculate experimental sample concentrations -------------------

# Using the standard curve just calculated, we can back-calculate the
# concentration of the standard concentrations, and more importantly
# the concentration of the experimental samples
plex_data <- plex_data %>%
```
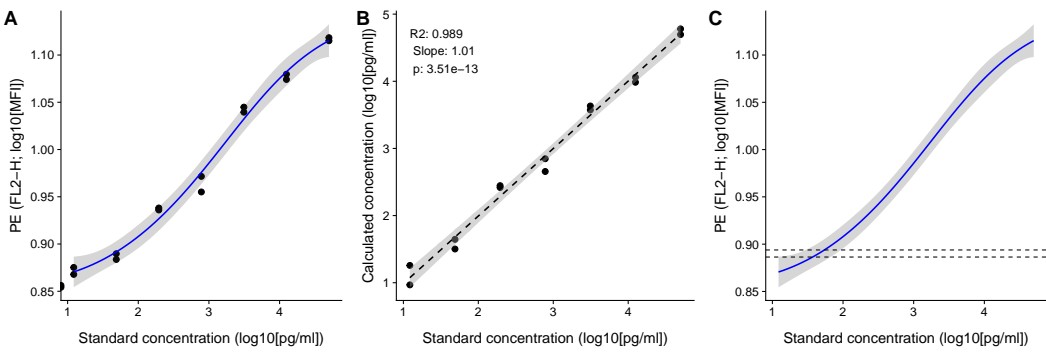

**Figure 4** **Visualization of standard and test samples for Angiopoietin-2.** The dataset included in `beadplexr` is from a 13-plex assay. Here we use Angiopoietin-2 to illustrate the visualizations. (A) A log–log plot of the standard curve of Angiopoietin-2. Each point is a single measurement (each in duplicate). The standard concentration is diluted in steps of four fold dilution from 50,000.0 to 12.21 pg/ml. The intensity of the analyte is measured in the PE channel. The full line indicates the best fit, and gray the confidence interval. (B) Correlation between the standard concentration ($x$-axis) and the calculated concentration of the standard samples ($y$-axis). The back calculation is done using the fit in (A) and the MFI of the samples. (C) Using the fit in (A) the concentration of an experimental sample is calculated. Visual inspection of the position of the experimental samples on the standard curve can reveal samples that are close to the upper or lower bound of the standard curve.

```
mutate('Standard data' =
    purrr::map2('Standard data', 'Model fit',
          calculate_concentration,
          .parameter = "FL2-H")) %>%
  mutate('Experimental data' =
      purrr::map2('Experimental data', 'Model fit',
            calculate_concentration, .parameter = "FL2-H"))

# Add concentration plots ------------------------------------

# We can also loop over each row and add plots to the data.frame
plex_data <- plex_data %>%
 mutate('Std curve' =
    purrr::pmap(list(.data = 'Standard data',
              .model = 'Model fit',
              .title = 'Analyte name'),
          plot_std_curve, .parameter = "FL2-H")) %>%
 mutate('Std conc' =
    purrr::map('Standard data',
          plot_target_est_conc)) %>%
 mutate('Est curve' =
    purrr::pmap(list('Experimental data',
              'Standard data',
```

```
                                    'Model fit',
                                    'Analyte name),
                              plot_estimate, .parameter = "FL2-H"))
```

Lastly we fulfill the purpose of all the previous actions and extract the concentration of each analyte for each sample.

```
# Extract analyte concentration --------------------------------

plex_data %>%
 unnest('Experimental data') %>%
 # Make the names a little more telling and transform them back to
 # useful concentrations
 rename('Concentration (pg/ml)' = Calc.conc,
     'Concentration error' = 'Calc.conc error') %>%
 mutate('Concentration (pg/ml)' = 10^ 'Concentration (pg/ml)',
     'Concentration error' = 10^ 'Concentration error')
```

## DISCUSSION

Multiplex bead assays make simultaneous evaluation of several analytes possible. Because of this, they are an attractive alternative to the commonly used sandwich ELISA. Commercial systems are available for acquisition on a standard flow cytometer, but these commercial systems make use of their own proprietary software for the data analysis. This can impose different limitations to the analysis. The R-package beadplexr, released under the MIT license, is meant as an open-source alternative to these commercial systems. The package is available from CRAN and from https://gitlab.com/ustervbo/beadplexr.

A critical step in the analysis multiplex bead assays is the identification of bead populations corresponding to each analyte. A single function in beadplexr acts as an interface to several common, and tested, clustering functions, making it easy to find the best suited clustering function. Future versions of the package will see improvements in this part, with inclusion of other clustering methods and perhaps a heuristic for automatic method selection.

Flow cytometry data are inherently noisy. beadplexr only provides a rudimentary function for removing points with no neighbors and lets the clustering functions determine which events are considered noisy though the .trim argument. However, a very noisy data set might make it difficult for an optimal identification of the bead clusters in the first place. De-noising multidimensional data is not trivial, but work is planned in this direction for a future release.

## CONCLUSION

The R-package beadplexr provides a frame work for easy and reproducible analysis of multiplex bead assays for the experienced and the novice user alike.

## ACKNOWLEDGEMENTS

The authors wish to thank Miltenyi Biotec, Bergisch Gladbach, Germany and C McGuckin, CTIBIOTECH, Lyon, France for the example data to test the package. We further acknowledge the support from the German Research Foundation (DFG) and the Open Access Publication Fund of Charité—Universitätsmedizin Berlin.

### Funding

This work was supported by BMBF grant e:KID. We received support from the German Research Foundation (DFG) and the Open Access Publication Fund of Charité – Universitätsmedizin Berlin. The funders had no role in study design, data collection and analysis, decision to publish, or preparation of the manuscript.

### Grant Disclosures

The following grant information was disclosed by the authors:
BMBF.
German Research Foundation (DFG).
Charité—Universitätsmedizin Berlin.

### Competing Interests

The authors declare there are no competing interests.

### Author Contributions

- Ulrik Stervbo conceived and designed the experiments, performed the experiments, analyzed the data, contributed reagents/materials/analysis tools, prepared figures and/or tables, authored or reviewed drafts of the paper, approved the final draft.
- Timm H. Westhoff and Nina Babel conceived and designed the experiments, authored or reviewed drafts of the paper, approved the final draft.

### Data Availability

Code and data is available from GitLab:

https://gitlab.com/ustervbo/beadplexr and from CRAN: https://CRAN.R-project.org/package=beadplexr.

### Supplemental Information

Supplemental information for this article can be found online at http://dx.doi.org/10.7717/peerj.5794#supplemental-information.

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
