# Peer review of "beadplexr: reproducible and automated analysis of multiplex bead assays"

_PeerJ, doi:10.7717/peerj.5794_

## Round 0.1 · original submission · Major Revisions

The authors presented an open source R package for the analysis of multiplex bead assays. The work has been appreciated by the referees although there are some major points to be addressed.

Reviewer 1 ·

Basic reporting

no comment

Experimental design

The authors address the need of having open environment for integrative analysis of data from different instruments.

I think there is a little need of automated population detection in common beads assays but I understand that manual gating in R is cumbersome and clustering is a viable alternative.
However applying clustering algorithm on every single sample is not very transparent or efficient. Also since the default algorithm "clara" uses downsampling such approach is not entirely reproducible.

Validity of the findings

no comment

Reviewer 2 ·

Basic reporting

* Overall, this paper needs much more thorough editing as there are many unclear sentences and grammar mistakes. For example the sentences starting on line 50, 108, 109 were hard for me to understand. Line 306 should include the word ‘clustering’ rather than ‘cluster’. I found too many such examples to list explicitly.
* The paper should provide more background and context, and cite both the commercial tools currently available for multiplex bead assay analysis, as well as the most similar open source tools for analyzing other types of bioassays, such as cellHTS and bioassayR. If any of those tools can be used for multiplex bead assay data the methods should be briefly compared and contrasted with the methods used here.
* It was mentioned that the examples analyze unpublished data. This data should be included in the R package, or swapped out for already published data, to make this work reproducible. Ideally, the full code and data to produce the examples and plots in the paper should be provided as a supplement, or as part of the documentation for the package.
* Figures 2-4 should include some context: (1) what data is visualized here? (2) What scientific interpretation or information can be learned from these plots?
* Throughout the paper, R functions in this package and other packages are referenced by name only. These should include a description of what the function does, and why it is used. It is essential that users be able to understand why each step was performed in the analysis workflows they perform based on this paper. For example, the use of ‘clara()’ on line 121.
* Please provide more detail on the meaning of ‘forward and side scatter properties’ as used on line 40.
* It found it very helpful for potential users that example R code is included directly in the manuscript, however the code examples don’t have enough context to be understood from the manuscript alone. I suggest adding code comments or a detailed description describing the commands in more detail, including the purpose of each option or setting when non-obvious.
* Since this package makes use of other Bioconductor packages (e.g. flowCore), have the authors considered distributing this via Bioconductor rather than CRAN and possibly providing more integration with related Bioconductor packages? This is just a suggestion, and should not be a requirement for publication.
* All R packages mentioned in the paper should be cited as references.

Experimental design

It was not clear exactly what statistical methods are used for the analysis here, and what the scientific justification for these methods is. The authors should include at least one new paragraph describing the scientific/mathematical purpose of each analysis step, how parameters are chosen, and justifying why this method is used over alternatives. Are any alternative statistical methods ever appropriate here?

Validity of the findings

The paragraph starting on line 119 mentions that finding the right clustering method is 'a matter of trial and error.’ Can the authors clarify why this is, for example explain why making a rigorous choice based on understanding the properties of each method in relation to the experiment would be fundamentally impossible? When doing a trial and error approach, how should the user/reader evaluate the performance and results of one method over another?

Additional comments

The authors report an open source R package for the analysis of multiplex bead assays, which fills a need within the scientific community. The package appears to be well designed, and includes documentation, examples, and unit tests. Substantial work is needed to clarify the manuscript in a way that will make it easy to understand for readers and potential users. Additionally, the paper seems to present an analysis of novel experimental data that is not explained in detail, nor provided to the user, which violates PeerJ guidelines. This data should be provided and explained, or else replaced with a different dataset.

As a reviewer, I would like to clarify that I have not used bead assays personally, so I can't speak to how well this package meets the specific needs of experimentalists performing these assays.

·

Basic reporting

Literature should be referenced appropriately throughout the paper, for instance Lines 23-24 would benefit from a reference to ELISA. Please make sure to reference the other multiplex systems for lines 33-39.

Experimental design

In line 52, the authors have mentioned that the other available software allows analysis of only a limited number of beads. I believe that this article would benefit from getting additional details on each of the available software and compare that to beadplexr in a tabular form highlighting the contributions of this R package that are perhaps lacking in the others.

Also, what is the maximum number of samples that can be analyzed at a single time and how long does it take to load the FACS file?

Validity of the findings

In the line 108, thought I understand that it would take a bit of trying to get the settings correct, but I believe that the paper would benefit from clarifying or rephrasing this sentence.

In the line 127, perhaps we could have an example to support for cases where mclust() is a better option.

Additional comments

I commend the authors for creating an open source tool for automated analysis of multiple bead assays. In addition clear commands to follow a workflow were also given. However, I wonder if the authors have considered creating a GUI for this tool using the R package Shiny https://shiny.rstudio.com/ so this package could become more user friendly.

---

## Round 0.2 · accepted · Accept

The paper has been amended and an acceptable clarification of the "Unpublished data" statement has been included by the authors. The paper is worthy the publication within PeerJ

# ·

Basic reporting

No comments

Experimental design

No comments

Validity of the findings

No Comments

Additional comments

I am satisfied by the changes made to the manuscript after the revisions.